# Moisture as a key factor alleviating low-temperature stress: Effects of hydrothermal conditions on maize emergence

Yang HuiYing [1,2], Gao Pan[1,2], Xu YingYing[1,2], Zhang GongLiang[1,2], Zheng Xu[3], Jiang Yu[4,5], M.I. Gang[4,5], Wang YuXian[1,2]*

1 Qiqihar Branch of Heilongjiang Academy of Agricultural Sciences, Qiqihar, China, 2 National Agricultural Experimental Station for Agricultural Environment, Qiqihar, China, 3 Heilongjiang University, Harbin, China, 4 Heihe Branch of Heilongjiang Academy of Agricultural Sciences, Heihe, China, 5 National Agricultural Experimental Station for Soil Quality, Heihe, China

* 13836209470@163.com

## Abstract

Early spring sowing of maize in semi-arid, wind-eroded regions is increasingly threatened by cold snaps due to climate change.These events, often coupled with uneven soil moisture distribution,compromise seedling emergence and early development. Identifying critical temperature and moisture thresholds is essential to ensure successful germination in these vulnerable environments.A factorial experiment was conducted in a controlled environment using maize seeds (Zea mays L.) exposed to diurnal temperature cycles.Treatments included five minimum temperatures (0,2,4,6,8°C), three chilling durations (2,4,6 hours),and four soil moisture levels (60,70,80,90% field capacity). Key germination metrics,including final germination rate, weighted germination time,synchrony,delay days,and seedling dry matter at day 30,were measured and analyzed using three-way ANOVA and Pearson correlations. Temperatures below 6°C significantly delayed germination and reduced final germination rates,particularly under low moisture conditions.Moisture levels ≥80% effectively mitigated chilling effects at moderate temperatures(4~6°C).Extended chilling durations further suppressed germination.The strongest interaction was observed between minimum temperature and soil moisture.Seedling dry matter accumulation was also significantly affected by all three factors and their interactions.Soil moisture serves as a critical buffer against chilling stress during maize germination. This study provides quantitative benchmarks for temperature and moisture combinations that optimize early maize emergence under extreme spring weather, offering practical insights for precision moisture management in semi-arid agriculture.

## Introduction

Over recent decades, early spring climatic variability has intensified globally, with increasing occurrences of late spring coldness, freeze–thaw cycles, and abrupt

**Data availability statement:** All relevant data are within the paper and its Supporting Information files.

**Funding:** The author(s) received no specific funding for this work.

**Competing interests:** The authors have declared that no competing interests exist.

cold snaps caused by climate change [1,2].These sudden thermal fluctuations pose significant risks to early-sown maize (Zea mays L.),especially in semi-arid, wind-eroded agricultural regions such as Northeast China, where chilling injury and stand loss have become increasingly frequent [3,4]. When soil temperatures drop below 10°C after sowing, seeds may suffer from imbibitional chilling injury, delayed emergence, and asynchronous stand establishment, often resulting in yield losses of up to 20~50% [5,6].

Temperature and soil moisture are the two primary abiotic factors that regulate seed germination and early seedling growth. The hydrothermal time (HTT) model suggests that germination is jointly driven by thermal time and water potential [7,8]. While optimal maize germination generally occurs between 15~30°C, temperatures below 10°C hinder enzymatic activity and delay radicle emergence [9,10].Simultaneously, water stress—particularly when water potential drops below –0.5 MPa—restricts oxygen diffusion and impairs metabolic activation, exacerbating the risk of failed emergence [11,12]. In field conditions, cold and dry soils can prolong emergence by 1~2 weeks and reduce germination rates by 30~60% [13,14].

Previous studies have mostly explored the independent effects of low temperature or soil moisture on seed germination; however, the interactive effects under early spring variable climate remain poorly understood [15,16]. Moreover, most studies overlook chilling duration, which significantly influences membrane stability, hormone balance, and oxidative stress responses [17,18]. Most existing studies are based on constant laboratory conditions and do not capture the transient and compound stresses of early spring sowing in continental semi-arid zones.This research gap is especially prominent in semi-arid regions, where climatic unpredictability is high. Without comprehensive models integrating temperature, moisture, and chilling duration, predictive accuracy for safe sowing is limited.

Emerging evidence suggests that adequate soil moisture may buffer chilling-induced damage by maintaining cellular hydration, preserving membrane integrity, and activating antioxidant systems and hormonal pathways [19–21]. However, this buffering capacity may decline with prolonged chilling or under extreme cold and saturated conditions [22,23]. Quantifying the hydrothermal thresholds governing safe emergence under varying chilling durations is thus critical for risk reduction and precise management.

To address these gaps, this study examines the interactive effects of chilling temperature, duration, and soil moisture on maize seed germination dynamics and early biomass accumulation. We hypothesize that (1) sufficient soil moisture alleviates chilling damage under moderate conditions; (2) prolonged chilling reduces this buffering effect; and (3) hydrothermal thresholds for safe emergence can be defined. Our results will contribute to improved early-sowing strategies and provide theoretical support for climate-resilient maize production. In the Qiqihar region, early spring sowing often coincides with snowmelt and occasional rainfall, which can temporarily raise soil moisture close to or above field capacity. This climatic context provides a practical background for interpreting the laboratory-based moisture treatments applied in this study.

## Materials and methods

### Experimental conditions

This study was conducted in a controlled-environment growth chamber at the Qiqihar Branch of the Heilongjiang Academy of Agricultural Sciences. The objective was to examine how low-temperature stress, duration of chilling, and soil moisture levels affect maize germination and early seedling growth. All experiments were carried out using a randomized complete block design with a full factorial arrangement of treatments.

### Experimental design

Three experimental factors were considered: minimum temperature (Tmin), chilling duration, and soil moisture content. Tmin levels were set at 0°C,2°C,4°C,6°C,and 8°C, while chilling durations were 2,4,and 6 hours. A control treatment (CK) maintained a constant temperature of 20°C. Soil moisture was set at four levels:60%,70%,80%, and90% of field capacity. We did not include 100% field capacity, because preliminary trials indicated that full saturation under low temperature often created transient waterlogging and hypoxic microenvironments, which do not reflect the well-aerated soil conditions of semi-arid spring seedbeds. Thus, 60–90% FC was selected to represent a practical range from sub-optimal to near-optimal moisture conditions while avoiding artefacts caused by excessive saturation. Each treatment was replicated three times.

### Plant material and substrate

The growth substrate consisted of fine sand that was autoclaved at 121 essive saturation.,d had an average particle size of 0.2e sand that was autoclaved at 121 essive saturation., which do not rH was determined to be 6.8 and the electrical conductivity 0.14 dS m21 essive saturation., which do not reflect the well-aerated soil conditions of semfe (core sampler) method, was 25% (w/w).

### Temperature simulation protocol

A six-phase diurnal temperature cycle was applied to mimic early-spring day–night thermal rhythms: Phase 1 (T1) $\rightarrow$ Tmin$\rightarrow$Phase 1 (T1) $\rightarrow$ Phase 2 (T2) $\rightarrow$ Tmax$\rightarrow$Phase 2 (T2), where Tmax was maintained at 20°C. Phase 1 and Phase 2 temperatures were calculated as:

T1 = (Tmin+Tmax)/3, T2=2 × (Tmin+Tmax)/3. Each phase lasted for 4 hours, mimicking early spring day-night thermal rhythms. Photoperiod was maintained at 12 hours (6:00–18:00) with 500 µmol/m²/s light intensity.

### Plant material and substrate

The maize hybrid Xianyu 696, commonly cultivated in northeastern China, was used as the experimental material. Seeds were surface sterilized prior to sowing. A well-washed and sterilized fine sand medium (passed through a 200-mesh sieve) was used as the growth substrate. Each replicate included 90 seeds sown in trays, which were placed in the climate chamber for germination.

### Moisture management

Soil moisture was maintained at designated levels (60∼90% field capacity) using a gravimetric method. Water was added daily based on weight difference, using the following formula:

Water added (g) = (Target moisture content−Current moisture content) × Dry soil weight.

Moisture content was monitored with a digital soil moisture meter and adjusted accordingly to minimize variation.

The experimental water used is double distilled water, with electrical conductivity (EC) of 1.2–1.5 µS/cm, total dissolved solids (TDS) of 0.5–1.2 ppm, and total organic carbon (TOC) of<5 ppb.

 

## Germination and growth measurements

Germination was recorded daily from day 2 after sowing. A seed was considered germinated when the coleoptile broke through the sand surface. The following metrics were calculated:

- Final Germination Rate (%) = (Number of emerged seeds/ Total sown seeds) × 100

- Weighted Germination Time (WGT): average germination timing considering emergence distribution

- Germination Synchrony: proportion of germinated seeds within peak emergence window

- Germination Delay Days: days until the first seed emerged

On day 30, seedlings were harvested, and shoot dry weight was measured after oven-drying at 80°C for 72 hours.

## Data analysis

All data were processed using Microsoft Excel 2010 and statistically analyzed in R software (version 4.1.0). Three-way ANOVA was used to test the effects of Tmin, duration, and moisture and their interactions. Pearson correlation analyses were conducted to evaluate relationships among key germination traits.Before ANOVA, data were examined for normality (Shapiro–Wilk test) and homogeneity of variances (Levene's test), and all datasets met these assumptions ($p > 0.05$). Significance levels were set at $p < 0.05$.

## Results

### Effects of minimum temperature (Tmin) on germination timing and rate

S1 Fig A1 illustrates the germination dynamics of maize under varying minimum temperatures (Tmin), chilling durations, and soil moisture levels.Germination was severely delayed at Tmin≤2°C,with final emergence rates below 50%,especially under 60% and 90% field capacity.In contrast, Tmin at 6°C and 8°C accelerated germination,reaching over 80% emergence within 7～10 days. The chilling duration had a compounding suppressive effect—longer exposure (4～6 hours) further delayed germination onset and reduced total emergence.Moisture demonstrated a non-linear buffering effect:70～80% field capacity yielded the most favorable germination outcomes across Tmin levels.Overly dry (60%) or saturated (90%) conditions reduced emergence under both moderate and extreme chilling.

Low-temperature duration exerted an additive suppressive effect.When Tmin was moderate (4～6°C),extending the exposure from 2 h to 6 h consistently delayed the emergence onset and reduced the final emergence percentage.This effect was most pronounced under drier(60%) and over-wet (90%) soil conditions.

Soil moisture showed a nonlinear interaction with Tmin.The 70% and 80% field capacity treatments supported the highest and fastest emergence across Tmin gradients.Both water deficit (60%) and oversaturation (90%) conditions impaired emergence,especially under extreme low Tmin.In control treatments (CK), germination curves were steep and saturated, indicating optimal conditions.

The sigmoidal germination pattern was most evident under Tmin≥6°C combined with 70～80% moisture.Under suboptimal conditions (e.g.,Tmin≤2°C),curves flattened or failed to approach saturation, indicating prolonged lag phases and reduced germination vigor.

The germination index (GI) exhibited a progressive increase over time across all treatments, with more rapid accumulation under higher Tmin and longer chilling durations (S2 Fig A2).Under favorable conditions (Tmin≥6°C and Duration=CK),GI values approached their maximum earlier (Day 5～6),whereas lower Tmin (0～4°C) and extended chilling duration (48～144 h) resulted in delayed and slower accumulation.Moisture level modulated the GI dynamics significantly:at suboptimal temperatures (e.g.,Tmin=2°C),higher moisture (80～90%) notably accelerated the GI curve,particularly under shorter chilling durations.

Germination speed(GS),represented as daily emergence rate,displayed a typical unimodal temporal distribution(S3 Fig A3).Peak GS was achieved earlier under warm and short chilling conditions (Tmin≥6°C,Duration = 2～4 h),while cold or prolonged chilling delayed the peak and lowered the magnitude.GS curves were markedly sensitive to soil moisture levels:at Tmin = 4～6°C, higher moisture (80～90%) produced sharper and earlier peaks,whereas under 0～2°C,excessive moisture (90%) appeared to suppress GS.The interactions among Tmin,Duration,and Moisture shaped the overall germination rhythm,confirming their joint regulation on both germination rate and uniformity.

### Effects of chilling duration and soil moisture on germination

Chilling duration imposed an additive inhibitory effect on germination performance. Extending exposure from 2 h to 6 h significantly increased weighted germination time and germination delay, especially under low Tmin, reflecting cumulative metabolic disruption and membrane damage consistent with elevated MDA and electrolyte leakage reported by Zhang et al. [24]. High soil moisture (80–90% FC) partially alleviated these delays at moderate temperatures (4–6°C), but its buffering capacity diminished under severe cold (≤2°C). Excessive moisture could even become detrimental, likely because saturated, cold soils restrict oxygen diffusion and impair metabolic activity. Overall, chilling duration and soil moisture interact to regulate maize germination, with adequate moisture offering protection only within a limited temperature–duration range.

### Three-way interactions and mechanistic implications

Three-way interaction plots (S12–S15 Figs D1–D4) revealed critical thresholds and nonlinear dependencies.For instance,at Tmin = 4～6°C and chilling = 4 h,80% moisture ensured >85% germination, while at Tmin = 0°C, even 90% moisture failed to surpass 50% germination.The delay days and WGT demonstrated strong Tmin×Duration×Moisture interactions, suggesting hydrothermal thresholds beyond which germination physiology collapses.

These patterns imply a mechanistic cascade: chilling disrupts energy metabolism and ROS balance;duration intensifies membrane and hormonal imbalance;moisture modulates oxygen diffusion and antioxidative defenses.Only when all three conditions reach favorable ranges can physiological recovery and coordinated emergence occur.These insights support integrating hydrothermal modeling into early sowing strategies.

### Three-way interactions and mechanistic implications

S16 Fig F and S1 Table present the three-way effects of Tmin,duration,and soil moisture on seedling dry matter accumulation measured at 30 days post-sowing.Tmin, duration, and moisture all had significant main effects (p < 0.001),with Tmin showing the largest effect size.Dry weight was lowest under Tmin = 0～2°C and 60% moisture,but increased significantly with warmer Tmin and higher moisture.Notably,the interaction between Tmin and moisture was significant (p = 0.005),while the three-way interaction also reached significance (p = 0.003),highlighting that dry matter production is co-regulated by hydrothermal conditions during early development.

## Discussion

In recent years, to ensure grain production, the wind–sand semi-arid regions of China have gradually promoted the use of integrated water–fertilizer drip irrigation technology, effectively alleviating yield losses caused by drought. However, in these regions, the interactive effects of low temperature, chilling duration, and soil moisture on maize emergence after early-spring sowing have not been systematically investigated. This study comprehensively evaluates the three-way interaction among chilling temperature (Tmin), chilling duration, and soil moisture on maize germination performance and early seedling growth.

## Interactive effects of chilling temperature, duration, and soil moisture

This study provides clear evidence that maize germination and emergence are jointly regulated by chilling temperature, exposure duration, and soil moisture. High soil moisture (≥80% field capacity) demonstrated a pronounced buffering effect under short-term moderate chilling conditions (Tmin=4~6°C, duration ≤4 h), as reflected by higher final germination rates, reduced weighted germination time, and enhanced germination synchrony. Such buffering effects are likely linked to sustained imbibitional hydration, maintenance of plasma membrane fluidity, and reduced mechanical impedance to radicle protrusion [9,12,15]. Adequate water availability during the imbibition phase ensures rapid and uniform water uptake, minimizes mechanical strain on expanding tissues, and facilitates metabolic activation.

However, the benefits of high moisture were not universal.Under prolonged chilling (≥6 h) or extreme cold (Tmin≤2°C), the buffering effect diminished and, in some cases, reversed—resulting in aggravated chilling injury. Excessive soil water under low temperatures can exacerbate hypoxic stress, restrict aerobic respiration, and enhance the accumulation of reactive oxygen species (ROS), thereby accelerating membrane lipid peroxidation and programmed cell death [25–27]. These findings suggest that the moisture buffering effect is highly conditional, offering protection under mild–moderate chilling and short exposure durations but losing efficacy—or even becoming detrimental—under extreme or prolonged stress conditions. This aligns with earlier studies that highlight the nonlinear, context-dependent nature of temperature–moisture interactions in seed physiology [15,26,27].Because the sterilized fine sand was nearly inert with minimal nutrient content and neutral pH, observed germination responses can be attributed primarily to the manipulated temperature and moisture conditions rather than to chemical soil factors.

## Hydrothermal imbalance under high moisture and low temperature

The co-occurrence of low temperatures and excessive soil moisture frequently creates a hydrothermal imbalance that adversely affects germination performance. Consistent with this, we intentionally did not include 100% FC in our treatments, as full saturation under chilling would likely exacerbate hypoxia and confound the interpretation of moisture buffering effects.Chilling temperatures reduce enzymatic activity and slow metabolic reactivation, while excess moisture limits oxygen diffusion in the seed microenvironment, inducing localized hypoxia [28–30]. This dual constraint impairs mitochondrial respiration, delays ATP synthesis, and suppresses the biosynthesis of growth-promoting hormones such as gibberellins, ultimately causing uneven and delayed germination [31–33].

Our experimental results showed that under Tmin=0~2°C and soil moisture ≥90%, germination synchrony dropped markedly, and weighted germination time was prolonged by 2~4days compared with optimal hydrothermal conditions. This pattern is consistent with previous reports indicating that cold, waterlogged soils generate hypoxic microsites that not only hinder germination but also predispose seedlings to early root and crown diseases, further reducing stand uniformity [34–36]. The observed reduction in synchrony has direct agronomic implications, as asynchronous emergence can increase intra-specific competition and hinder uniform crop development, leading to uneven canopy closure and reduced yield potential.

## Moisture thresholds and weakening of buffering under severe cold

The buffering effect of soil moisture showed a clear decline with decreasing temperature and increasing chilling duration. When Tmin fell below 2°C and chilling duration exceeded 6 h, even high soil moisture failed to maintain critical physiological protective mechanisms, such as membrane stabilization and ROS detoxification [37–39]. This suggests the existence of a temperature–moisture–duration threshold, beyond which the benefits of high soil moisture sharply diminish.

The weakening of the buffering effect under extreme cold may be explained by several mechanisms: (i) cold-induced phase transitions in membrane lipids that reduce fluidity and permeability; (ii) inhibition of aquaporin-mediated water transport, slowing cellular rehydration; and (iii) prolonged oxidative stress causing irreversible damage to cellular macromolecules [22,40,41]. These processes ultimately disrupt metabolic homeostasis and reduce seed viability.

Identifying such thresholds is essential for developing precision sowing recommendations, especially in semi-arid, cold-prone regions where both soil moisture and temperature are highly variable in early spring. The quantitative hydrothermal interaction data generated in this study provide a valuable basis for threshold-based decision-support tools that can guide sowing time selection, irrigation scheduling, and seed treatment protocols. Incorporating these thresholds into predictive models could improve risk assessment and help farmers optimize emergence success under increasingly variable early-spring climates.

## Implications for biomass accumulation and agronomic practices

Our findings demonstrate that early-stage hydrothermal stress exerts prolonged effects on maize growth and biomass accumulation, with implications that extend well beyond the emergence phase. Dry matter accumulation at the seedling stage was significantly reduced when plants were exposed to severe chilling (Tmin=0~2°C) combined with low soil moisture (≤60% field capacity). This reduction can be attributed to restricted leaf area expansion, lower chlorophyll content, suppressed photosynthetic rate, and impaired root development, which collectively limit the plant's capacity to intercept light and acquire nutrients [19,42,43]. At the cellular level, early hydrothermal stress may reduce mesophyll conductance, inhibit Rubisco activation, and disrupt phloem loading, thereby constraining assimilate transport to growing tissues [44].

The consequences of reduced seedling biomass often persist into subsequent growth stages. In maize, a suboptimal early biomass status can delay the transition from vegetative to reproductive growth, reduce the rate of leaf area index (LAI) development, and limit the crop's ability to capture radiation during critical periods of kernel set [45–46]. This "carry-over effect" means that even if environmental conditions improve later in the season, yield potential may remain compromised due to reduced source capacity (photosynthate supply) and weakened sink strength (ear and kernel development) [47–49]. Additionally, stress-induced shifts in biomass allocation—such as a higher root-to-shoot ratio under early drought—may further influence water and nutrient uptake dynamics throughout the growing season [50].

Conversely, our results showed that under moderate chilling (Tmin=4~6°C) combined with high soil moisture (≥80% field capacity), seedling biomass was comparable to the control, suggesting the occurrence of physiological acclimation or "stress memory" effects. Such acclimation may involve sustained activation of antioxidant defense systems, osmotic adjustment, and the upregulation of protective proteins that mitigate subsequent stress impacts [51–52]. This finding highlights the potential for moisture management to partially buffer against chilling injury, particularly under conditions that do not exceed the physiological tolerance limits of maize seedlings.

From an agronomic perspective, maintaining soil moisture at or above 80% field capacity during early sowing is critical for ensuring both rapid, uniform emergence and robust early biomass accumulation. Adequate early biomass not only supports strong canopy development but also enhances competitive ability against weeds, improves resource-use efficiency, and provides resilience against mid-season stress events [53–54]. Integrating chilling temperature–duration–moisture interaction thresholds into sowing calendars and irrigation scheduling could greatly enhance the resilience and yield stability of maize production systems in semi-arid, cold-prone regions. Furthermore, incorporating these thresholds into breeding programs could guide the selection of genotypes with improved early-stage cold tolerance and water-use efficiency, thereby strengthening climate-resilient maize production strategies [55–56].

A limitation of this study is the use of sterilized sand as the growth medium, which lacks the complexity of field soils in terms of organic matter, microbial communities, and water retention, potentially limiting direct extrapolation to field conditions. Nevertheless, field surveys in the Qiqihar region indicate that most maize fields are sandy with low water and nutrient retention, so our use of sand reasonably simulates local conditions. Extending the experiment to later vegetative stages (e.g., V3 or V5) was not feasible in growth pots due to restricted root space and light availability, which may not accurately reflect field growth. Future studies could assess long-term recovery at later stages to better understand the full impact of chilling and soil moisture stress on maize growth. In addition, maize fields in Qiqihar are typically large, and a single planter can sow only 7–10 km$^2$ per day. To ensure all fields are sown within the optimal planting window, farmers often need to sow early, even under low temperature or high soil moisture conditions, to avoid delaying crop maturity. This

practical constraint helps explain why early sowing under suboptimal conditions is common, providing context for inter-preting the relevance of our experimental findings to field decision-making.

## Conclusions

This study systematically examined the interactive effects of chilling temperature (Tmin), chilling duration, and soil mois-ture on maize seed germination and early seedling growth under controlled diurnal temperature regimes. The results demonstrate that both the severity and duration of chilling stress markedly suppressed germination performance, particu-larly when soil moisture levels were suboptimal.

Soil moisture was identified as a critical buffering factor against chilling injury. Moisture levels maintained at 80~90% of field capacity effectively alleviated the adverse impacts of chilling stress, enhancing final germination rate, accelerating emergence, and improving synchrony, especially under moderate Tmin conditions (4~6°C). In contrast, under extreme chilling (0~2°C), elevated soil moisture was insufficient to fully counteract cold damage, particularly during prolonged exposure, indicating that the buffering effect of moisture is both temperature- and duration-dependent.

The three-way interaction among Tmin, chilling duration, and soil moisture significantly influenced all germination-related parameters, including weighted germination time, emergence delay, and synchrony, as well as early-stage seed-ling dry matter accumulation. These findings underscore the necessity of defining hydrothermal thresholds to guide optimal sowing decisions in regions sensitive to climatic variability.

To translate our quantitative findings into actionable agronomic advice, we propose a practical guideline for sowing decisions under variable spring conditions in Table 1. These recommendations are synthesized directly from our experi-mental thresholds on final germination rate, synchrony, and seedling dry matter accumulation.

These guidelines emphasize that soil moisture at or above 80% field capacity can create a safe sowing window under moderate cold stress. However, this buffering effect has limits and cannot overcome the physiological damage caused by severe chilling (Tmin ≤ 2°C). Farmers are advised to use short-term weather forecasts, particularly for minimum tempera-ture and its expected duration, in conjunction with soil moisture measurements to make risk-informed sowing decisions.

From an agronomic perspective, maintaining soil moisture at or above 80% of field capacity during early sowing is rec-ommended as an effective strategy to enhance stand establishment under "false spring" conditions. This practice may be especially beneficial in semi-arid regions increasingly affected by erratic spring temperature fluctuations.

This study provides quantitative benchmarks of hydrothermal interactions that may inform sowing strategies for early-spring maize under variable climate conditions. Future work will explore a wider range of minimum temperatures (including up to 12°C) to extend the applicability of these findings. Moreover, the findings establish a foundation for future research on the hormonal regulation of hydrothermal stress responses during germination, which will be the focus of sub-sequent investigations.

**Table 1. Practical guidelines for maize sowing decisions based on minimum temperature (Tmin) forecasts and manageable soil moisture levels.**

| Minimum Tempera-ture (Tmin) | Chilling Dura-tion Forecast | Recommended Soil Mois-ture (% Field Capacity) | Expected Emergence Outcome | Sowing Recommendation |
|---|---|---|---|---|
| Moderate Chilling (4–6°C) | Short (≤ 4 hours) | ≥ 80% | High germination rate (>85%), good synchrony, minimal delay | Safe to sow. Optimal conditions for stand establishment. |
| Moderate Chilling (4–6°C) | Prolonged (≥ 6 hours) | ≥ 80% | Good germination rate, poten-tial slight delay | Sow with caution. Prioritize fields where moisture can be maintained. |
| Severe Chilling (0–2°C) | Any duration | Any level | Poor germination rate (<50%), significant delay, low synchrony | Avoid sowing. High risk of stand failure. Wait for warmer conditions. |
| Fluctuating Spring Con-ditions (Unpredictable) | Variable | Maintain 70–80% | Variable, but risk of stand loss is reduced | Sow if irrigation is available. Moisture buffer is critical to mitigate unexpected cold snaps. |

## Supporting information

**S1 Fig A1. Cumulative Germination Curves under Varying Hydrothermal Conditions.** Cumulative germination (%) over time in maize seeds subjected to different minimum temperatures (Tmin: 0~8°C), chilling durations (2,4,6 h), and soil moisture levels (60%,70%,80%,90% field capacity).Each curve represents the mean of three replicates (n = 90), with error bars indicating standard deviation. The control (CK) group was maintained at 20°C for 24 h without chilling stress.
(TIFF)

**S2 Fig A2. Daily Germination Index Dynamics under Chilling Treatments.** Daily germination index (%) of maize under various chilling temperature, duration, and moisture combinations. Each point reflects the average value from three replicates. Lower Tmin and prolonged duration suppressed the index, while higher soil moisture (80~90%) accelerated accumulation under moderate cold stress.
(TIFF)

**S3 Fig A3. Germination Speed Dynamics in Response to Chilling and Moisture.** Temporal trends of daily germination speed (%) under varying chilling durations (2,4,6 h and CK) and soil moisture levels. Loess curves illustrate average emergence rates, with shaded areas representing 95% confidence intervals. Color indicates Tmin level.
(TIFF)

**S4 Fig B1. Weighted Germination Time (WGT) under Chilling and Moisture Conditions.** WGT values across treatments of Tmin, chilling duration, and soil moisture levels. Higher Tmin and moisture resulted in earlier average emergence timing. Error bars denote standard error (SE) of mean values across replicates.
(TIFF)

**S5 Fig B2. Final Germination Rate across Hydrothermal Treatments.** Final germination rate (%) as influenced by Tmin, chilling duration, and soil moisture content. Data are mean±SE(n = 3). Higher Tmin and moisture supported better germination under chilling exposure.
(TIFF)

**S6 Fig B3. Germination Synchrony under Combined Stress Conditions.** Synchrony (%) of maize germination under different Tmin, chilling durations, and soil moisture levels.Synchrony is calculated as the peak emergence proportion relative to total emergence. Values represent mean±SE.
(TIFF)

**S7 Fig B4. Germination Delay Days under Chilling and Soil Moisture Stress.** Mean number of days until first emergence under varying Tmin,duration,and soil moisture levels.Greater delays occurred at Tmin≤2°C and low moisture.Bars represent mean±SE.
(TIFF)

**S8 Fig C1. Effects of Soil Moisture on Final Germination Rate under Different Chilling Durations.** Final germination rate (%) of maize seeds under soil moisture levels of 60–90% field capacity and chilling durations of 2, 4, 6, and 24 h. Increased soil moisture significantly enhanced final germination rate across all chilling durations.
(TIFF)

**S9 Fig C2. Effects of Soil Moisture on Germination Synchrony under Different Chilling Durations.** Germination synchrony (%) of maize seeds at 60–90% field capacity subjected to chilling durations of 2, 4, 6, and 24 h. Higher soil moisture promoted more synchronized germination, particularly under prolonged chilling stress.
(TIFF)

**S10 Fig C3. Effects of Soil Moisture on Germination Delay under Different Chilling Durations.** Germination delay (days) of maize seeds under varying soil moisture conditions (60–90% field capacity) and chilling durations (2, 4, 6, and 24 h). Increasing soil moisture effectively reduced germination delay under all chilling treatments.
(TIFF)

**S11 Fig C4. Effects of Soil Moisture on Weighted Germination Time under Different Chilling Durations.** Weighted germination time (days) of maize seeds under soil moisture levels of 60–90% field capacity and chilling durations of 2, 4, 6, and 24 h. Elevated soil moisture shortened weighted germination time, indicating improved germination performance under chilling stress.
(TIFF)

**S12 Fig D1. Interactive Effects of Tmin, Chilling Duration, and Soil Moisture on Final Germination Rate.** Final germination rate (%) of maize seeds across combined treatments of minimum temperature (Tmin), chilling duration, and soil moisture. Lines represent means±SE. Correlation coefficients (R) and corresponding p values are shown in each panel to quantify the relationships among the interactive factors.
(TIFF)

**S13 Fig D2. Interactive Effects of Tmin, Chilling Duration, and Soil Moisture on Germination Synchrony.** Germination synchrony (%) of maize seeds under different combinations of Tmin, chilling duration, and soil moisture. Values are presented as means±SE. The strength and significance of correlations are indicated by R and p values displayed within each panel.
(TIFF)

**S14 Fig D3. Interactive Effects of Tmin, Chilling Duration, and Soil Moisture on Germination Delay.** Germination delay (days) of maize seeds in response to the interactive effects of Tmin, chilling duration, and soil moisture. Lines denote means±SE, with correlation analysis results (R and p values) provided in each panel.
(TIFF)

**S15 Fig D4. Interactive Effects of Tmin, Chilling Duration, and Soil Moisture on Weighted Germination Time.** Weighted germination time (days) of maize seeds across combinations of Tmin, chilling duration, and soil moisture. Data are shown as means±SE, and correlation coefficients (R) and p values are reported within each panel.
(TIFF)

**S16 Fig F. Dry Matter Accumulation under Interactive Chilling and Moisture Conditions.** Shoot dry weight (g plant$^{-1}$) of maize seedlings at 30 days after sowing. Points represent individual replicates; regression lines indicate trends across moisture levels under different Tmin and chilling durations. ANOVA summary provided in S1 Table.
(TIFF)

**S1 Table. Three-way ANOVA summary for the effects of Tmin, chilling duration, and soil moisture on maize shoot dry matter accumulation.** DFn=degrees of freedom numerator; DFd=degrees of freedom denominator;F=F-value; p=p-value; ges=generalized eta squared.Bolded p-values indicate statistical significance (p<0.05).
(PDF)

## Author contributions

**Conceptualization:** Huiying Yang, MI Gang.

**Data curation:** Huiying Yang, Zheng Xu, Jiang Yu, MI Gang.

**Formal analysis:** Huiying Yang.

**Investigation:** Huiying Yang, Gao Pan, Zheng Xu.

**Methodology:** Gao Pan, Zheng Xu, Jiang Yu.

**Project administration:** Wang YuXian.

**Resources:** Xu YingYing, Zheng Xu.

**Supervision:** Zheng Xu.

**Validation:** Zheng Xu.

**Visualization:** Zhang GongLiang.

**Writing – original draft:** Huiying Yang.

**Writing – review & editing:** Huiying Yang.

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
