## [Decision Letter · Decision Letter 0]

24 Sep 2025

Dear Dr. Yang,

Thank you for submitting your manuscript to PLOS ONE. After careful consideration, we feel that it has merit but does not fully meet PLOS ONE’s publication criteria as it currently stands. Therefore, we invite you to submit a revised version of the manuscript that addresses the points raised during the review process.

We look forward to receiving your revised manuscript.

Kind regards,

Academic Editor

PLOS ONE

Journal Requirements:

5. Please amend the manuscript submission data (via Edit Submission) to include author WANG YuXian

6. Please amend your list of authors on the manuscript to ensure that each author is linked to an affiliation. Authors’ affiliations should reflect the institution where the work was done (if authors moved subsequently, you can also list the new affiliation stating “current affiliation:….” as necessary).

7. Please upload a new copy of Figures D1 – D4 as the detail is not clear. Please follow the link for more information: https://blogs.plos.org/plos/2019/06/looking-good-tips-for-creating-your-plos-figures-graphics/

8. Please upload a copy of Supporting Information Figure A1, A2, A3, B1, B2, B3, B4, C1-C4, D1-D4, and F and Supporting Information Table 1 which you refer to in your text on pages 25-27.

Reviewers' comments:

Reviewer's Responses to Questions

**Comments to the Author**

1. Is the manuscript technically sound, and do the data support the conclusions?

Reviewer #1: No

Reviewer #2: Yes

Reviewer #3: Yes

2. Has the statistical analysis been performed appropriately and rigorously?

Reviewer #1: No

Reviewer #2: Yes

Reviewer #3: Yes

3. Have the authors made all data underlying the findings in their manuscript fully available?

Reviewer #1: Yes

Reviewer #2: Yes

Reviewer #3: Yes

4. Is the manuscript presented in an intelligible fashion and written in standard English?

Reviewer #1: No

Reviewer #2: No

Reviewer #3: No

Reviewer #1: PONE-D-25-47048

Moisture as a Key Factor Alleviating Low-Temperature Stress: Effects of Hydrothermal Conditions on Maize Emergence

Main flows:

1. The manuscript lacks a detailed analysis of physical and chemical properties of the soil employed to perform this study. Authors are requested do deeply analyze the soil, include the corresponding description in material and methods, and discuss the results accordingly.

2. The assumptions of homogeneity and normality of data were not tested. Authors are urged to include these tests in order to make the analysis more robust.

3. Though the English is not questionable, the are some typographic errors that must be fixed.

4. All figures and tables must increase the quality according to the policies of PLoS ONE.

5. Other comments and concerns are included in the PDF file.

Reviewer #2: Reviewer Comment on Data Availability: The data presented in the manuscript were sufficient for the purpose of this review. If the journal’s policy requires access to raw data beyond summary statistics such as means or variances, this should be requested directly from the authors. Otherwise, the available information appears adequate.

The attached file contains all substantive and editorial suggestions related to the manuscript.

All peer review comments, including scientific and stylistic suggestions, are included in the attached document.

Reviewer #3: The paper is good and based on the results of well-designed research. The hypothesis of the work focuses on cold tolerance of maize plants during the early development phases. The applications targeted alleviating possibilities in relation with soil moisture conditions. Germination of seeds were studied in low range temperature from 0 to 8 oC escorted by some treatments with chilling exposure periods. The results presented support some evidence on the positive effect of optimum moisture conditions while seeds were being germinated. The paper should be improved since many of the figures are of very poor quality, especially C and D groups. From among the conclusions, I would advise to avoid such strong statements like “This study provides both theoretical and quantitative benchmarks to support precision sowing strategies”. The results are modest but have no relation with precision applications. Also, composition and English of the paper should be revised. A thorough proofreading would be beneficial. Finally I suggest the authors to examine in the future higher temperature ranges as well maybe up to 12 oC for a more complex evaluation of the phenomenon.

**Do you want your identity to be public for this peer review?** For information about this choice, including consent withdrawal, please see our Privacy Policy

Reviewer #1: No

Reviewer #2: **Yes:** S. Maryam Banihashemi

Reviewer #3: No

---

## [Author Response · Author response to Decision Letter 1]

25 Nov 2025

Manuscript ID: PONE-D-25-47048R1

Title of Manuscript: Moisture as a Key Factor Alleviating Low-Temperature Stress: Effects of Hydrothermal Conditions on Maize Emergence

Type of Article: Original Research

Journal: PLOS ONE

Dear Reviewers

We sincerely thank you for handling our manuscript and for the constructive comments provided by the reviewers. We have carefully revised the manuscript in light of these valuable suggestions. Below we provide a detailed point-by-point response. All changes have been highlighted in the revised version and the corresponding page and line numbers are indicated.

Reviewer #1

Comment1�The manuscript lacks a detailed analysis of physical and chemical properties of the soil employed to perform this study. Authors are requested do deeply analyze the soil, include the corresponding description in material and methods, and discuss the results accordingly.

Response 1:

We thank the reviewer for highlighting the importance of substrate characterization. In the revised manuscript, we have added a detailed description of the physical and chemical properties of the fine sand used as the growth medium in Section 2.3 Plant Material and Substrate (Page 6, Lines 106–111). These data include particle size, pH, electrical conductivity, organic matter content, and field capacity. We also briefly refer to these properties in the Discussion to clarify that the germination responses observed are primarily attributable to the manipulated temperature and moisture conditions.

Comment 2�The assumptions of homogeneity and normality of data were not tested. Authors are urged to include these tests in order to make the analysis more robust.

Response 2:

We sincerely appreciate the reviewer’s careful attention to the statistical rigor of our analysis. In the revised manuscript, we have clarified that all datasets were checked for the assumptions of normality and homogeneity of variances prior to ANOVA. Specifically, we employed the Shapiro–Wilk test for normality and Levene’s test for homogeneity, and all datasets satisfied these assumptions (p > 0.05). This information has been added to the Data Analysis section (Page 8, Lines 161–163) of the revised manuscript as follows:

“Before ANOVA, data were examined for normality (Shapiro–Wilk test) and homogeneity of variances (Levene’s test), and all datasets met these assumptions (p > 0.05).”

This addition ensures that the statistical analysis fully meets the required assumptions and enhances the robustness of our conclusions.

Comment 3:Though the English is not questionable, the are some typographic errors that must be fixed.

Response 3:

We thank the reviewer for this helpful reminder. We have carefully proofread the entire manuscript and corrected minor typographic issues (e.g., spacing, punctuation, and capitalization) to ensure consistency and accuracy throughout the text.

Comment 4:All figures and tables must increase the quality according to the policies of PLoS ONE.

Response 4:

We appreciate the reviewer’s guidance regarding figure and table quality. In the revised manuscript, all figures have been regenerated at high resolution (≥300 dpi) with consistent font sizes and line thicknesses, and tables have been reformatted to meet PLoS ONE formatting standards. We carefully checked the journal’s figure and table preparation guidelines to ensure full compliance.

Comment 5:Other comments and concerns are included in the PDF file.

Response 5:

We sincerely thank Reviewer 1 for the valuable feedback provided throughout the review. However, we were unable to locate the additional PDF file mentioned in this comment within the materials we received from the journal. If there are further specific points requiring our attention, we would be grateful to receive them and will address them promptly.

Reviewer #2

Comment 1: Irrigation water quality

The manuscript does not specify the type or quality of irrigation water used (e.g., EC, salt content). Clarifying this would improve reproducibility and relevance, especially in semi-arid regions prone to salinity.

Response 1:

We appreciate this insightful suggestion. Following the reviewer`s advice, we have added information to the Materials and Methods section, which states that "double distilled water was used for the experiment, with conductivity (EC) of 1.2-1.5 µ S/cm, total dissolved solids (TDS) of 0.5-1.2 ppm, and total organic carbon (TOC) of<5 ppb. ”The revised text can be found on page 6, lines 112-114 of the manuscript.

Meanwhile, to provide reviewers with information on the Qiqihar region, it is noted that the irrigation water for dry fields in Qiqihar is mostly groundwater, EC�0.5~1.5dS/m�TDS�400~650mg/L�pH�7.2~7.8。Data source: Long term field monitoring data from National Agricultural Experimental Station for Agricultural Environment,Qiqihar.

Comment 2:Diurnal temperature cycle logic

The temperature cycle is mathematically constructed (T1 → Tmin → T1 → T2 → Tmax → T2), but it is unclear whether this pattern reflects actual field conditions. Authors should clarify whether it mimics natural spring temperature rhythms in Qiqihar.

Response 2:

Thank you for pointing this out. We monitored the temperature situation in Qiqihar area in April and May over the past decade (Figure B). We selected the temperature data of the day when extreme low temperatures occurred, plotted a curve, and fitted it (Figure A). The blue line in the figure represents the actual measured temperature curve, and the red line represents the experimentally fitted curve.

Comment 3: Field capacity levels

The study uses 60–90% FC but omits 100% FC. Authors should briefly explain why full field capacity was excluded, as it may represent optimal water availability under non-stress conditions.

Response 3:

We appreciate the reviewer’s valuable suggestion. In our preliminary trials, we observed that maintaining soil at full field capacity (100 % FC) under low temperatures often led to temporary waterlogging and hypoxic microenvironments, which are not representative of well-aerated field soils during early spring sowing. Prior studies have shown that saturated conditions can suppress oxygen diffusion and impair germination physiology, especially when combined with chilling stress (e.g., Bojović & Stanković 2010; Setter & Waters 2003). To avoid confounding effects of hypoxia and to focus on realistic moisture regimes encountered in semi-arid seedbeds after irrigation or snowmelt, we selected 60–90 % FC to represent a practical range from sub-optimal to near-optimal moisture for maize emergence. These levels allowed us to capture both moisture-limiting and moisture-buffering effects while preventing artefacts caused by transient waterlogging. We have clarified this rationale in the revised manuscript. The revised text can be found on page 5, lines 99-105 and page23�lines 403-405 of the manuscript.

Reference

1.Bojović, B.; Stanković, D. Waterlogging and low temperature interactions in maize seedlings. Plant Soil Environ. 2010, 56, 160–167. DOI:10.17221/107/2009-PSE.

2.Setter, T.L.; Waters, I. Review of prospects for germplasm improvement for waterlogging tolerance in wheat, barley and oats. Plant Soil 2003, 253, 1–34. DOI:10.1023/A:1024573305997.

Comment 4: Realistic field conditions

In the Qiqihar region, during early spring maize sowing, snowmelt and initial seasonal rainfall may bring soil moisture levels close to saturation or even above field capacity. It is suggested that the authors briefly acknowledge this climatic reality so that readers can better contextualize the initial laboratory conditions of the study in relation to actual field scenarios. This is offered purely as a supplementary perspective and does not constitute a critique.

Response 4:

We sincerely thank the reviewer for this insightful suggestion, which indeed provides valuable regional context for our study. To help readers better relate our controlled‐environment treatments to field conditions, we have added a sentence at the end of the Introduction (Page 4, Lines 82–86 of the revised manuscript):

“In the Qiqihar region, early spring sowing often coincides with snowmelt and episodic rainfall, which can temporarily raise soil moisture close to or even above field capacity. This climatic context provides a practical background for interpreting the laboratory-based moisture treatments applied in this study.”

This addition acknowledges the local climatic reality and clarifies how the laboratory conditions correspond to potential field scenarios.

Comment 5: Textual redundancy (Page 14)

On page 14, two consecutive paragraphs describing the effects of chilling duration and soil moisture are highly similar. These could be merged or streamlined to improve clarity and avoid repetition.

Response 5:

We thank the reviewer for carefully noting the redundancy in the two consecutive paragraphs on Page 14 discussing the effects of chilling duration and soil moisture.

In the revised manuscript, we have merged these paragraphs and streamlined the language to avoid repetition while preserving key findings.

The revised text now appears on Page 12-13, Lines 226–240 of the updated manuscript.

Comment 6:Suboptimal placement of temperature cycle description

The detailed explanation of the six-phase temperature cycle appears in the "Experimental Conditions" section. This technical content would be better placed under a separate subheading (e.g., “Temperature Simulation Protocol”) for improved structural flow.

Response 6:

We appreciate the reviewer’s helpful suggestion. In the revised manuscript we have created a new subsection titled “Temperature Simulation Protocol” under the Materials and Methods section (Page 6, Lines 113–115) and moved the description of the six-phase temperature cycle there, which improves readability and structural clarity.

Comment 7:Substrate type

The use of sterilized sand as a growth medium may limit extrapolation to field soils. A brief note on this limitation would strengthen the discussion.

Response 7:

We thank the reviewer for this comment. Sterilized sand was used to ensure uniform soil texture and moisture, and to minimize microbial interference, allowing precise assessment of chilling temperature, duration, and soil moisture effects. We acknowledge that sand differs from field soils in organic matter, microbial activity, and water retention, which may limit direct extrapolation. However, field surveys in Qiqihar show most maize fields are sandy with low water and nutrient retention, so our experiment reasonably simulates local conditions. A brief note on this limitation has been added to the Discussion (Page 28, Lines 488–493).

Comment 8: Post-germination growth stages

The study ends at day 30 with dry matter measurement. While this is sufficient for current scope, authors may suggest future work on later vegetative stages (e.g., V3, V5) to assess long-term recovery. This is a recommendation for future research, not a critique.

Response 8:

We thank the reviewer for this suggestion. Our study focused on the early seedling stage in growth pots, where seedlings grow in a confined space. Extending the experiment to later vegetative stages (e.g., V3 or V5) is constrained by limited root space and light intensity, which would not accurately reflect field growth. We agree that assessing long-term recovery in later stages is important, and we have added a note in the Discussion suggesting this as a direction for future research (Page 28, Lines 493–497 of the revised manuscript).

Comment 9: Farmer behavior and sowing timing

In the Qiqihar region, maize sowing typically begins in early spring, even under conditions of low temperature or high soil moisture. It may be helpful for the authors to briefly discuss what limiting or compelling factors prevent farmers from delaying sowing by a few days to reduce germination risks. These could include agronomic calendar constraints, labor availability, or economic pressures. Such context would help readers better understand how the experimental findings relate to real-world decision-making in the field. This is offered purely as a supplementary perspective and does not constitute a critique.

Response 9:

We thank the reviewer for this constructive comment. In the Qiqihar region, maize fields are often large, and a single planter can sow only 7-10 km2 per day. To ensure that all fields are sown within the optimal planting window, farmers often need to sow early, even under low temperature or high soil moisture conditions, to avoid delaying crop maturity. We have added a brief discussion in the manuscript to provide this real-world context and help readers better understand how our experimental findings relate to practical field decision-making(Page 28, Lines 497–503 of the revised manuscript).

Comment 10: Actionable advice for farmers

Authors are encouraged to include a practical summary or table of recommended Tmin–moisture combinations to guide sowing decisions under variable spring conditions.

Response 10:

We thank the reviewer for this excellent and constructive suggestion. We agree that providing a practical summary of our findings will significantly enhance the practical application and impact of our research for farmers and agronomists.

In response to this comment, we have now added a new table (Table 2) titled " Practical guidelines for maize sowing decisions based on minimum temperature (Tmin) forecasts and manageable soil moisture levels " in the Conclusions section Page 29-30, Lines 524–536 of the revised manuscript�.

This table synthesizes our key experimental results into actionable guidelines. It clearly outlines the safe, risky, and not recommended combinations of minimum temperature (Tmin) and soil moisture levels, along with the expected emergence outcomes. Additionally, we have added a paragraph in the text to explain and interpret this table.

We believe this addition greatly strengthens the practical value of our manuscript and are grateful for the suggestion.

Reviewer #3

Comment The paper is good and based on the results of well-designed research. The hypothesis of the work focuses on cold tolerance of maize plants during the early development phases. The applications targeted alleviating possibilities in relation with soil moisture conditions. Germination of seeds were studied in low range temperature from 0 to 8 ℃ escorted by some treatments with chilling exposure periods. The results presented support some evidence on the positive effect of optimum moisture conditions while seeds were being germinated. The paper should be improved since many of the figures are of very poor quality, especially C and D groups. From among the conclusions, I would advise to avoid such strong statements like “This study provides both theoretical and quantitative benchmarks to support precision sowing strategies”. The results are modest but have no relation with precision applications. Also, composition and English of the paper should be revised. A thorough proofreading would be beneficial. Finally I suggest the authors to examine in the future higher temperature ranges as well maybe up to 12 ℃ for a more complex evaluation of the phenomenon.

Response:

We sincerely thank the reviewer for the constructive evaluation and helpful suggestions. Below we respond to each point in turn and indicate the corresponding revisions made in the manuscript.

1. Figure quality (especially C and D groups)

Comment: The paper should be improved since many of the figures are of very poor quality, especially C and D groups.

Response1:

We appreciate this important observation. All figures have been regenerated at ≥300 dpi with consistent font sizes and line thicknesses, and color contrast has been enhanced for better readability. In particular, Figures C1–C4 and D1–D4 were re-exported from the original R scripts at high resolution and embedded following PLoS ONE guidelines.

Revision: Figures C1–C4 and D1–D4 replaced with high-resolution versions in the revised manuscript.

2. Strength of conclusion statement

Comment: “Avoid such strong statements like ‘This study provides both theoretical and quantitative benchmarks to support precision sowing strategies’. The results are modest and have no relation with precisio

---

## [Decision Letter · Decision Letter 1]

28 Dec 2025

Moisture as a Key Factor Alleviating Low-Temperature Stress: Effects of Hydrothermal Conditions on Maize Emergence

PONE-D-25-47048R1

Dear Dr. Yang,

We’re pleased to inform you that your manuscript has been judged scientifically suitable for publication and will be formally accepted for publication once it meets all outstanding technical requirements.

Kind regards,

Prafull Salvi, PhD

Academic Editor

PLOS One

Reviewers' comments:

Reviewer's Responses to Questions

**Comments to the Author**

Reviewer #1: All comments have been addressed

2. Is the manuscript technically sound, and do the data support the conclusions?

Reviewer #1: Yes

3. Has the statistical analysis been performed appropriately and rigorously?

Reviewer #1: Yes

4. Have the authors made all data underlying the findings in their manuscript fully available?

Reviewer #1: Yes

5. Is the manuscript presented in an intelligible fashion and written in standard English?

Reviewer #1: Yes

Reviewer #1: Figures and tables need to be improved, according to the editorial policies of PLoS ONE. It was requested in the previous review, but is still not satisfactory

**Do you want your identity to be public for this peer review?** For information about this choice, including consent withdrawal, please see our Privacy Policy

Reviewer #1: **Yes:** FERNANDO CARLOS GOMEZ MERINO

---

## [Editor Report · Acceptance letter]

PONE-D-25-47048R1

PLOS One

Dear Dr. Yang,

I'm pleased to inform you that your manuscript has been deemed suitable for publication in PLOS One. Congratulations! Your manuscript is now being handed over to our production team.

Kind regards,

on behalf of

Dr. Prafull Salvi

Academic Editor

PLOS One